# High HER2 Intratumoral Heterogeneity Is a Predictive Factor for Poor Prognosis in Early-Stage and Locally Advanced HER2-Positive Breast Cancer

**DOI:** 10.3390/cancers16051062

**Published:** 2024-03-05

**Authors:** Tomonori Tanei, Shigeto Seno, Yoshiaki Sota, Takaaki Hatano, Yuri Kitahara, Kaori Abe, Nanae Masunaga, Masami Tsukabe, Tetsuhiro Yoshinami, Tomohiro Miyake, Masafumi Shimoda, Hideo Matsuda, Kenzo Shimazu

**Affiliations:** 1Department of Breast and Endocrine Surgery, Graduate School of Medicine, Osaka University, 2-2-E10 Yamadaoka, Suita 565-0871, Osaka, Japan; y_sota123@onsurg.med.osaka-u.ac.jp (Y.S.);y.kitahara@onsurg.med.osaka-u.ac.jp (Y.K.); abe216@onsurg.med.osaka-u.ac.jp (K.A.); mtsukabe@onsurg.med.osaka-u.ac.jp (M.T.); yosinami-te@onsurg.med.osaka-u.ac.jp (T.Y.); t_miyake@onsurg.med.osaka-u.ac.jp (T.M.); mshimoda@onsurg.med.osaka-u.ac.jp (M.S.);; 2Department of Bioinformatic Engineering, Graduate School of Information Science and Technology, Osaka University, 1-5 Yamadaoka, Suita 565-0871, Osaka, Japan; senoo@ist.osaka-u.ac.jp (S.S.); matsuda@ist.osaka-u.ac.jp (H.M.)

**Keywords:** breast carcinoma, intratumoral heterogeneity, HER2 gene expression, prognosis, breast neoplasms

## Abstract

**Simple Summary:**

Breast cancer tumors are considered to have intratumoral, as well as human epidermal growth factor receptor 2 (HER2), heterogeneity. Tumors with high intratumoral heterogeneity (ITH) have demonstrated therapeutic resistance. However, studies on cancer heterogeneity as a prognostic factor in breast cancer have been limited. Therefore, we evaluated HER2 ITH, which was manifested by the shape of HER2 fluorescence in situ hybridization amplification distributed histograms (HER2 FISH distributions) with the HER2 gene copy number within a tumor sample. We aimed to determine whether high HER2 heterogeneity is clinically significant for poor prognosis due to resistance to postoperative adjuvant therapy with HER2-targeted agents in primary breast cancer. Indeed, we were able to show herein that high HER2 heterogeneity is significantly associated with poorer prognosis in patients with HER2-positive breast cancer. These results indicated that high HER2 heterogeneity is a factor predicting poor prognosis in patients with HER2-positive breast cancer. Our present observation seemed to be clinically important, because it is expected that our HER2 FISH distribution analysis of heterogeneity might be a convenient and clinically useful method for the prognosis prediction of patients after HER2 adjuvant therapy.

**Abstract:**

Purpose: Breast cancer tumors frequently have intratumoral heterogeneity (ITH). Tumors with high ITH cause therapeutic resistance and have human epidermal growth factor receptor 2 (HER2) heterogeneity in response to HER2-targeted therapies. This study aimed to investigate whether high HER2 heterogeneity levels were clinically related to a poor prognosis for HER2-targeted adjuvant therapy resistance in primary breast cancers. Methods: This study included patients with primary breast cancer (*n* = 251) treated with adjuvant HER2-targeted therapies. HER2 heterogeneity was manifested by the shape of HER2 fluorescence in situ hybridization amplification (FISH) distributed histograms with the HER2 gene copy number within a tumor sample. Each tumor was classified into a biphasic grade graph (high heterogeneity [HH]) group or a monophasic grade graph (low heterogeneity [LH]) group based on heterogeneity. Both groups were evaluated for disease-free survival (DFS) and overall survival (OS) for a median of ten years of annual follow-up. Results: Of 251 patients with HER2-positive breast cancer, 46 (18.3%) and 205 (81.7%) were classified into the HH and LH groups, respectively. The HH group had more distant metastases and a poorer prognosis than the LH group (DFS: *p* < 0.001 (HH:63% vs. LH:91% at 10 years) and for the OS: *p* = 0.012 (HH:78% vs. LH:95% at 10 years). Conclusions: High HER2 heterogeneity is a poor prognostic factor in patients with HER2-positive breast cancer. A novel approach to heterogeneity, which is manifested by the shape of HER2 FISH distributions, might be clinically useful in the prognosis prediction of patients after HER2 adjuvant therapy.

## 1. Introduction

Cancer heterogeneity refers to the diversity of cancer cells within a tumor as well as the differences between tumors in different individuals [1,2]. It is a fundamental characteristic of cancer and can have important implications for diagnosis, treatment, and prognosis. Tumor heterogeneity includes genetic mutations, epigenetic changes, and cellular level signaling pathway differences [3,4]. This causes subpopulations of cancer cells within a tumor with different characteristics, such as different growth rates, treatment responses, and metastasis potential [5]. The presence of heterogeneity can make developing effective cancer therapies challenging, as effective treatments against one cancer cell subpopulation may be ineffective against others. Breast cancer tumors have intratumoral heterogeneity (ITH) and tumors with high ITH have demonstrated therapeutic resistance [6]. Also, human epidermal growth factor receptor 2 (HER2) expression can be heterogeneous in breast cancer tumors. HER2 is a tyrosine kinase receptor which is encoded by the ERBB2 gene as a cancerogenic gene. HER2 is an orphan receptor characterized by a constitutively activated conformation and lacks specific ligands or ligand binding activity. The formation of HER2 homodimers leads to the phosphorylation of the tyrosine kinase domain, thereby activating various downstream cancer signaling pathways, ultimately enhancing cancer cell proliferation, tumor formation, invasion, and related processes [7,8].

HER2-positive breast cancer is more aggressive than other breast cancer types, but it can be treated with targeted therapies such as trastuzumab and pertuzumab, which specifically target the HER2 protein [9,10]. Overall, the prognosis of patients with HER2-positive early breast cancer has significantly improved in recent years due to advances in targeted therapies. Currently, the primary application of HER2 assessment lies in the improvement of patient survival to anti-HER2 therapy in the neoadjuvant and adjuvant settings. HER2 heterogeneity is defined by the coexistence of at least two distinct cellular clones with differing HER2 statuses within the same tumor, and the heterogeneity is the coexistence of different tumor cell subpopulations with varying levels of HER2 protein expression [11]. The definition of HER2-positive cancer is typically determined primarily with immunohistochemistry (IHC) and in situ hybridization (ISH). ISH involves the simultaneous detection of HER2 gene amplification and chromosome 17 (CEP17) to assess HER2 gene status in tumor tissue. If the IHC test reveals a score of 3+, indicating strong HER2 protein expression, or if the IHC test shows a score of 2+ along with confirmed HER2 gene amplification via ISH, the cancer is classified as HER2-positive [12]. HER2 heterogeneity is a well-established phenomenon in breast cancer, signifying the potential for distinct regions within the same tumor to exhibit varying levels of protein expression or genetic amplification [7,11]. Tumor heterogeneity refers to the diversity of cancer cells within a tumor or among tumors, encompassing variations in genetic mutations, gene expression patterns, and cellular characteristics. This heterogeneity can occur spatially, with different regions of a tumor exhibiting distinct features, and temporally, as cancer cells evolve and adapt over time [2,13,14]. Heterogeneous HER2 amplification was observed in 11–40% of tumor cell populations in HER2-positive breast cancers [8,15,16,17]. On the geographic (spatial) distribution, HER2 heterogeneity has been described in a “clustered” form or “mosaic” pattern characterized by pockets of highly amplified cells [8,18,19,20]. HER2 heterogeneity may contribute to inaccurate HER2 status assessment and affect therapeutic decision-making, and the validation of techniques to identify HER2 heterogeneity in the clinic and the concurrent development of agents to effectively treat tumors with nonuniform HER2 expression are needed [21,22]. Dual-color in situ hybridization (DISH) is also a molecular diagnostic technique used to assess the status of the HER2 gene in breast cancer and other malignancies [23,24,25]. DISH probes are designed to bind to the HER2 gene and CEP17 and can be visualized under a microscope. DISH allows for the simultaneous detection of both the HER2 gene and CEP17, providing comprehensive information on gene amplification, and allows for quantitative analysis, enabling the precise measurement of gene amplification levels.

Recently, Seol et al. reported that the presence of HER2 heterogeneity, associated with decreased disease-free survival (DFS), was treated with only 25% adjuvant trastuzumab therapy [26]. In contrast, Shen et al. revealed that HER2 heterogeneity, detected with a HER2 gene protein assay (GPA), was associated with poor overall survival (OS) and increased distal metastasis in patients with HER2-positive breast cancer [21,27]. In the report, HER2 gene amplification was evaluated with a HER2 GPA, which combines HER2 IHC and DISH for a simultaneous evaluation of both the protein and gene at the tissue level. However, The DISH method frequently presents with non-specific background deposition, including debris of subtle silver deposits, which elude automated analyses. Furthermore, the DISH method yields manual counting dependent on the measurer, requiring training for signal counting and making it susceptible to inter-measurer variability in interpretation [23,28]. Recently, Laurence et.al. investigated the prevalence of a range of gene expression distributions in three different tumor types from the Cancer Genome Atlas (TCGA) and studied a gene distribution shape to model the heterogeneity of transcriptomic data [29]. Also, their results indicated that prognostic genes identified based on the consideration of the shape of the distribution were different from those identified through more standard assumptions. Histogram analysis is the most common and popular method of characterization of ITH in imaging data [30]. Radziuviene et al. demonstrated the automated image analysis of HER2 FISH histograms to refine their definition of genetic heterogeneity and found that HER2 ITH could also be identified by histograms based on HER2 FISH amplification [31]. It is necessity to develop novel methods capable of automatically analyzing HER2 ITH within cancer tissues and to make unbiased analyses, rather than relying on manual procedures. Also, to assess more clear-cut heterogeneities of HER2, a novel method is necessary to evaluate the HER2 copy number quantification, as HER2 gene copy numbers are present in each of those cancer cells. In addition, it is desirable that the method proves beneficial in predicting prognosis and anti-HER2 therapy sensitivity. We developed a novel method to confirm the shape of HER2 FISH distributions with the HER2 gene copy number within a tumor sample and automatically analyzed HER2 ITH. Also, we implemented our method to investigate the association between HER2-targeted therapy and HER2 heterogeneity. This study defined HER2 ITH using a Gaussian mixture modeling (GMM) analysis of histograms based on HER2 FISH distributions. Recently, a GMM classifier demonstrated the potential use for the clinical validation of markers and determination of target populations to improve the molecular stratification of patients with breast cancer [32].

We aimed to develop a novel method to automatically analyze HER2 ITH and determine the clinical association of high HER2 heterogeneity with a poor prognosis due to resistance to postoperative adjuvant therapies with HER2-targeted agents in primary breast cancer, through the application of GMM analysis.

## 2. Materials and Methods

### 2.1. Patient Selection and Breast Tumor Samples

This study consecutively recruited 251 patients with HER2-positive invasive breast carcinoma (mean age: 56.6 years; range: 28–97 years) treated with adjuvant or neoadjuvant HER2-targeted therapies. These patients underwent mastectomy or breast-conserving surgery from February 2009 to January 2018 at Osaka University Hospital, Osaka, Japan. All 251 patients received 12 months of adjuvant trastuzumab.

Surgery was performed on 185 patients, followed by anti-HER2 adjuvant chemotherapy, and 66 patients received neoadjuvant chemotherapy followed by surgery (anthracycline and taxane-based chemotherapy, *n* = 146; taxane-based chemotherapy, *n* = 87; anthracycline chemotherapy, *n* = 5). Both anti-HER2 adjuvant therapy and hormonal therapy were given to 162 patients with hormone-receptor-positive cancer (aromatase inhibitors, *n* = 99; tamoxifen, *n* = 45; or goserelin + tamoxifen, *n* = 18). We collected clinical–pathological characteristics from patients’ electronic medical records, and Table 1 summarizes the details. Tumor tissues (surgical specimens) were obtained at surgery, and tumor tissues from patients who received neoadjuvant chemotherapy were obtained before the treatment by a vacuum-assisted core needle biopsy. Solid intratumoral tissues were fixed in 10% buffered formalin and embedded in paraffin (FFPE). The Ethics Committee of Osaka University approved the study protocol.

### 2.2. HER2 FISH Assay

All tumor tissues underwent a HER2 FISH assay. HER2 status, including HER2 gene signals and ratios, was determined by FISH using a PathVysion HER-2 DNA probe kit (Abbott Laboratories, Abbott Park, IL, USA). Regarding the kits, the tumor FFPE tissue slides were dried, and a 20 µL DAPI counterstain solution was applied. Then, a fluorescence microscope was used to capture both orange–fluorescent signals corresponding to HER2/neu and green–fluorescent signals corresponding to CEP17. For the histograms of HER2 FISH, all of the fluorescence images were automatically identified by histograms based on HER2 FISH amplification using the protocol of the Duet-3/Setup Station/SOLO2 system (BioView, Rehovot, Israel). Meanwhile, for counting the values of HER2 FISH, a fluorescence microscope was used to count the number of those fluorescent signals under a manual or semi-automated protocols analysis. First, samples were analyzed with a semi-automated protocol, and then the samples that failed the semi-automated protocol analysis were analyzed with a manual protocol. The number of cells counted was more than 60 in the semi-automated protocols, and it was 20 in the manual ones. The ratio between the HER2/neu and CEP17 signal counts was calculated. HER2 was considered positive when a tumor contained more than two genes per cell for the FISH assay of HER2. Experienced molecular pathologists used the 2018 HER2 ASCO/CAP updated guidelines to analyze the HER2 status.

### 2.3. Assessment of HER2 Intratumoral Heterogeneity

We evaluated HER2 intratumoral heterogeneity (ITH), which was manifested by the shape of HER2 FISH distributions with the HER2 gene copy number within a tumor sample, as shown in Figure 1A. We used scanned image data from FISH diagnostic reports. We extracted the FISH count information by reading the histogram and performing an image processing analysis, such as OCR, because a few scanned FISH images were insufficient for image analysis. Then, the heterogeneity was analyzed by a Gaussian mixture modeling (GMM) of the histograms based on HER2 FISH. Parameters were estimated by fitting a mixed distribution with mean μ → σ, and π was the weight of each mixed component:*f*(*x*) = π_1_*N*(*x*∣μ_1_, σ_1_) + π_2_*N*(*x*∣μ_2_, σ_2_)

This equation shows the Gaussian distribution *f*(*x*) to the distribution of signal counts, where *N* is the normal and standard deviation.

Here, the distributions of FISH signal counts of 2 and those >2 are assumed to be normal (π→, respectively). Then, 1/2 was used as a measure to calculate the diversity. This is because a large 1/2 means a FISH count of <2, but the average FISH count was >2. Appendix A shows two representative cases where imaging for a diagnostic report was fitted with a mixture of Gaussian distributions. HER2 ITH, which was manifested by the shape of HER2 FISH distributions, was classified into two groups: the biphasic grade graph (high heterogeneity [HH]) group and the monophasic grade graph (low heterogeneity [LH]) group for each tumor in Figure 1B. Both groups were evaluated in terms of DFS and OS. All patients had been receiving HER2 treatment. A median of 10 years after annual follow-ups was obtained. We used Python (v.3.10.6) and the Python libraries “opencv” (v.4.7.0.72), “pillow” (v.9.0.1), “pyOCR” (v.0.8.3), and “pytesseract” (v.0.3.10) for diagnostic report image processing.

Figure 1 shows the representative histograms of HER2 FISH distributions.

### 2.4. Statistical Analysis

All patients were classified into the HH and LH groups, and the HH groups were compared to the LH groups in clinicopathological parameters. The chi-square test was used to assess the associations between HER2 heterogeneity and clinicopathological parameters. DFS was calculated from the date of diagnosis of primary breast cancer to the date of diagnosis of metastatic breast cancer, and OS was calculated from the date of diagnosis of metastatic breast cancer to the date of death from any cause, censoring at the last known follow-up. The Kaplan–Meier method was used to calculate t disease-free survival (DFS) rates. Differences in DFS were evaluated by the log-rank test. The Cox proportional hazard model was used for the univariate and multivariate analyses of various clinicopathological and biological factors for DFS. R (v.4.2.3; 14) was used for all statistical analyses, and statistical significance was set to *p* < 0.05 for two-sided tests.

### 2.5. Data Availability

We obtained Medical Ethical Committee approval (ethical board approval number of Osaka University: 22080 (T1)). This was an observational study conducted with patient enrollment based on the opt-out method for comprehensive informed consent. Informed consent was obtained in the form of opt-out on a website, and those who rejected were excluded.

## 3. Results

### 3.1. Association between HER2 Intratumoral Heterogeneity and Patient Characteristics

HER2-positive breast cancer was classified into the HH and LH groups with high (18.3%) and low (81.7%) cases, respectively, of 251 tumors. The subsequent analysis considered HER2-positive breast cancers showing a biphasic- or monophasic-grade graph analysis of the HER2 FISH distributions as the HH or LH groups, respectively. The HER2-positive breast cancer tumors within the HH group had a tendency toward a higher proportion of HER2 IHC 2+ than 3+ expression, and the HER2 FISH ratio in the HH group was notably expressed higher than that of the LH group (Table 2). The tumors within the HH group were significantly associated with estrogen receptor (ER) + (*p* = 0.002) and PgR + (*p* < 0.001). No significant association was observed between HER2 heterogeneity (high or low) and menopausal status, histological type, histological grade, lymph node status, or stage. Appendix A shows the frequency of HH HER2-positive tumors following the ER status and progesterone estrogen receptor (PgR). The ER+ and PgR+ tumors demonstrated the highest frequency of HH HER2 heterogeneity (ER+ and PgR+: 28% [28/99], ER− and PgR−: 8% [6/75], ER+ and PgR−: 6% [4/70], and ER− and PgR+: 0% [0/7]).

### 3.2. HER2 Intratumoral Heterogeneity and Patient Prognosis

The DFS and OS rates of patients with tumors in the HH group were significantly longer and shorter, respectively (DFS: *p* < 0.001 (HH group: 63% vs. LH group: 91% at 10 years), and OS: *p* = 0.012 (HH group: 78% vs. LH group: 95% at 10 years)) (Figure 2A,B). A univariate analysis of various prognostic factors demonstrated that pretreatment lymph node metastasis and resected lymph node metastasis were significantly associated with patient prognosis (DFS, pretreatment: *p* = 0.029, resected: *p* = 0.009; OS, pretreatment: *p* = 0.014, resected: *p* = 0.009) (Table 3 and Table 4). A multivariate analysis revealed that HER2 heterogeneity was significantly associated with prognosis (DFS: *p* < 0.001, OS: *p* = 0.039). High levels of HER2 ITH were clinically significant for poor prognosis for resistance to adjuvant therapy with HER2-targeted therapies in primary breast cancers (Appendix A). The ratio of HER2 FISH (the mean number of HER2 FISH signals/cell) of the HH group (median: 2.3) was lower than that of the LH group (median: 5.4). However, the ratio of HER2 FISH was not significantly associated with prognosis. It was necessary to investigate whether the prognosis of LH was associated with the level of HER2 FISH amplification. LH tumors were divided into the ex-high amplification LH group (Ex-High LH) (>4.9%) and high amplification group LH (High LH) (<4.9%) according to the median value of HER2 FISH as the cut-off, and we investigated whether the relationship of HER2 heterogeneity with prognosis was associated with the ratio of HER2 FISH (Figure 1B). LH tumors in the Ex-High LH group and High LH group were not associated with prognosis (DFS: *p* = 0.253, OS: *p* = 0.686) (Figure 2C,D). Appendix A show the representative results between patient recurrence and the ratio of HER2 FISH. In particular, the proportion (percentage) of patient recurrence in the HH group was higher than that in the LH group.

Figure 2 shows disease-free survival (DFS) and overall survival (OS) rates according to HER2 intratumoral heterogeneity.

## 4. Discussion

This study aimed to develop a novel method to automatically analyze HER2 intratumoral heterogeneity (ITH) and to clarify the clinicopathological characteristics and prognosis of patients with breast cancer with HER2 ITH (Appendix A). We revealed an 18.3% frequency of HH tumors. HH tumors were significantly associated with high ER positivity, PgR positivity, and a lower HER2 FISH ratio than LH tumors. Thus, luminal HER2 (ER+ and PgR+) tumors had the highest frequency (28%) of being HH tumors (Appendix A) in our study, and these observations were essentially consistent with the concept of tumor heterogeneity, where a biological diversity of cancer cells is found within a tumor [33]. In Table 2, HH tumors were not significantly associated in pretreatment LN metastasis (0.098). The reason was that HER2 ITH may contribute to the therapeutic effect (the suppression of metastatic recurrence) of HER2 treatment and may have little to do with cancer invasion, metastasis, and the progression to LN metastasis. High proliferative cancer cells in both HH and LH tumors may be susceptible to pretreatment LN metastasis.

Further, the present study demonstrated a difference in DFS and OS rates between patients with HH and LH tumors, indicating that the presence of high heterogeneity has prognostic significance (Table 3). High heterogeneity was an independent prognostic factor after adjustment for other factors, including the type of anti-HER2 adjuvant chemotherapy. Highly heterogeneous breast cancers may show poorer prognosis because of their association with different biological characteristics. On the other hand, the ratio of HER2 FISH and HER2 IHC (3+, or 2+) could not predict prognosis in patients with HER2-positive breast cancer. In addition, we investigated whether the level of HER2 FISH amplification was associated with the prognosis of LH, and LH tumors were divided into the ex-high amplification LH group (Ex-High LH) and high amplification group LH (High LH). As a result, in HER2-positive LH tumors, our study revealed that the ratio of HER2 FISH (the mean number of HER2 FISH signals/cell) (Ex-High LH vs. High LH) was not associated with the prognosis of DFS rates or OS rates (Figure 2C,D).

Similarly, Shen et al. reported a significantly poorer prognosis in patients with HH tumors than in those with LH tumors, which was evaluated with HER2 gene protein assay (GPA), which combined HER2 IHC (the protein) and DISH (HER2 gene amplification) at the tissue level [21,27]. In addition, Hou et al. and Horii et al. revealed that HER2 ITH with the HER2 GPA method was an independent factor predicting an incomplete response to anti-HER2 neoadjuvant chemotherapy [34,35]. However, the HER2 GPA method as well as DISH yields manual counting and demands advanced technical expertise and specialized equipment. Their methods lacked standardized staining and interpretation methods, making it susceptible to false positives influenced by background factors such as debris. Also, the HER2 GPA was not analyzed with the distribution of the HER2 gene copy number within a tumor sample.

The strength of our method was that it was able to automatically analyze HER2 ITH within cancer tissues and to make unbiased analyses of HER2 ITH, rather than relying on manual procedures. Also, our method could evaluate the HER2 copy number quantification within a tumor sample and assess more clear-cut heterogeneities of HER2, because HER2 gene copy numbers are present in each of those cancer cells within a tumor sample. An automated analysis of HER2 expression typically involves utilizing computer algorithms and image processing techniques to assess the HER2 IHC status in tissue samples. Miglietta et.al. reported that the assessment of HER2 IHC status had significant observer variability depending on the pathologist [18]. To accurately assess the state of HER2 IHC, alternative methods such as molecular examinations and digital pathology have been proposed [36]. Since the widespread adoption of whole-slide imaging, digital image analysis has emerged as a swift, cost-effective, objective, and reproducible scoring methodology for evaluating HER2 IHC status and the efficacy of anti-HER2 therapies through immunohistochemistry [37,38,39,40]. However, a digital image analysis for evaluating the heterogeneity of HER2 IHC, which correlates with anti-HER2 treatment efficacy, is yet to be developed. Our study had the limitations of not comparing HER2 IHC with HER2 gene copy numbers within a tumor sample, unlike HER2 dual-color imaging, which combines HER2 IHC and HER2 gene amplification. However, we believe that the prognosis in patients of HER2-positive breast cancers can be estimated more accurately by adding our HER2 FISH distribution analysis of heterogeneity.

Moreover, another strength of our analysis was that it was an easy, low-cost method that only read histograms of FISH diagnostic reports and performed an image processing analysis. The findings of this study provide valuable insights into the efficacy of our novel method of HER2 ITH analysis and highlight its potential as a clinically useful method to predict prognosis and anti-HER2 therapy sensitivity. Moreover, the clinical significance of the identification of our analysis to predict prognosis seems to deserve further investigation with multicenter research.

Furthermore, our analysis had the potential to accurately assess ITH not only in HER2-positive breast cancers but also in HER2-low ones. It was hypothesized that HER2-low breast cancers commonly represent a more heterogeneous population than HER2-positive breast cancers [41]. Recently, trastuzumab deruxtecan (T-DXd) treatment resulted in significantly longer relapse-free survival and OS than chemotherapy among patients with HER2-low metastatic breast cancer [42]. This study reported that HER2-low tumors constituted a heterogeneous population that varied in prognosis and sensitivity to systemic treatments. T-DXd effectively targets tumor cells that express low levels of HER2 and can deliver a potent cytotoxic payload through the bystander effect to neighboring tumor cells heterogeneously expressing HER2, unlike many other approved HER2-targeted therapies [43,44]. Therefore, in the HER2-low as well as HER2-positive, highly heterogeneous breast cancers might be responsive to the treatment of T-DXd that exhibited potent therapeutic efficacy through the delivery of a cytotoxic payload. In addition, Mosele et al. reported that HER2 was a determinant of sensitivity to T-DXd, although modest anti-tumor activity was also observed in a small subset of patients whose cancer did not express HER2 [45]. The study revealed that T-DXd anti-tumor activity increased when HER2 expression was high; modest anti-tumor activity was also observed in patients with HER2 IHC 0 and suggested that very low levels of HER2 could allow an uptake of T-DXd and/or partially mediated drug efficacy. That is to say, if the level of HER2 IHC is extremely low or absent, it cannot have implications for T-DXd treatment decisions, and our HER2 FISH distribution analysis without HER2 IHC might accurately identify the heterogeneity of HER2-low breast cancer or absence thereof and predict T-DXd efficacy on breast cancers. Hence, our analysis may be useful for predicting the effectiveness of not only standard HER2-targeted therapies but also of this new drug against highly heterogeneous tumors. In the future, we will investigate whether our method can accurately identify the heterogeneity of HER2-low breast cancers or not.

Meanwhile, it has been confirmed that HER2-positive gastric cancer responds to HER2-targeted therapy as well [46]. The positivity rate of HER2 in gastric cancer is lower compared to that in breast cancer, resulting in the lower effectiveness of HER2-targeted therapies [47,48]. Moreover, there is a higher probability of HER2 heterogeneity in gastric cancer compared to that in breast cancer [49,50]. Our study might have a clinical practice impact, in that our novel analysis clinically has the potential to become a new classification method of HER2 ITH and predict prognosis in patients with HER2-positive gastric cancer as well as breast cancer, as a discriminative, inexpensive, and easy-to-perform method.

## 5. Conclusions

We have revealed the HH of HER2 to be a poor prognostic factor in patients with HER2-positive breast cancer. Examining HER2 heterogeneity using HER2 FISH distributions is a discriminative, inexpensive, and easy-to-perform method. Our results indicate that our HER2 FISH distribution analysis of HER2 ITH might be clinically useful in the prognosis prediction of patients after HER2 adjuvant therapy.

## Figures and Tables

**Figure 1 cancers-16-01062-f001:**
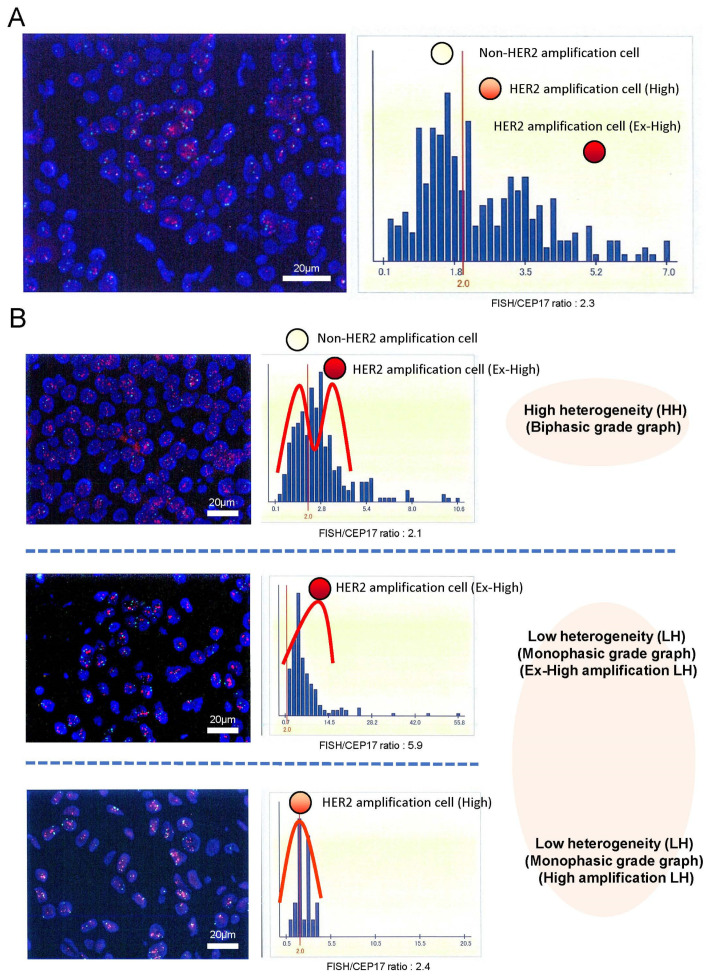
HER2 intratumoral heterogeneity was identified by the shape of HER2 FISH amplification distributed histograms with the HER2 gene copy number within a tumor sample (**A**). Those histograms were classified into two groups: the biphasic grade graph (high heterogeneous [HH] group) and the monophasic biphasic grade graph (low heterogeneous [LH] group) magnification ×400 (**B**). Additionally, the LH group was divided into the ex-high amplification LH (Ex-High LH) group and the high amplification LH (High LH) group by the ratio of HER2 FISH (mean number of HER2 FISH signals/cell). Representative images on the left were HER2 FISH imaging with HER2 (red signals)/CEP17 (green signals) in tumor cell nuclei (DAPI: blue) magnification ×400.

**Figure 2 cancers-16-01062-f002:**
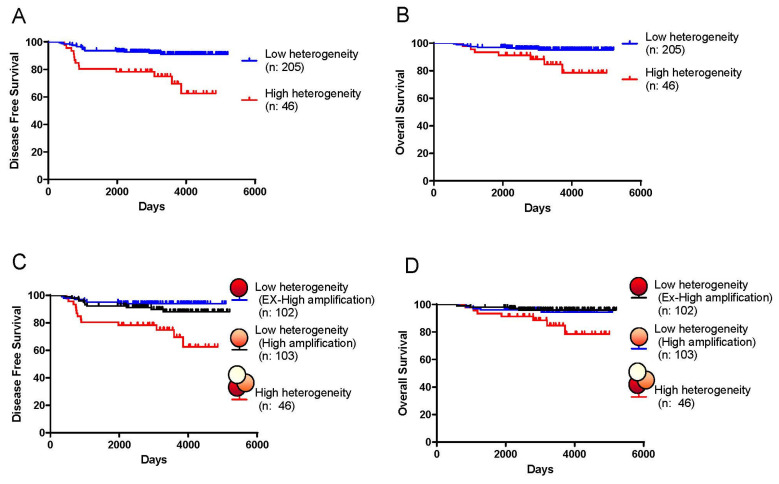
Disease-free survival (DFS) and overall survival (OS) rates according to HER2 intratumoral heterogeneity. DFS (**A**) and OS rates (**B**) of patients with breast cancer comparing the high heterogeneous (HH) group to the low heterogeneous (LH) group (DFS: *p* < 0.001 (HH group: 63% vs. LH group: 91% at 10 years), and OS: *p* = 0.012 (HH group: 78% vs. LH group: 95% at 10 years)). DFS rates (**C**) and OS rates (**D**) of patients with breast cancer were compared between the HH, Ex-High LH, and High LH groups.

**Table 1 cancers-16-01062-t001:** Patient characteristics.

	Number	(%)		Number	(%)
Menopause			ER		
Pre-	96	(38.2)	Positive	167	(66.5)
Post-	155	(61.8)	Negative	84	(33.5)
Histological type			PgR		
IDC	244	(97.2)	Positive	99	(39.4)
ILC	7	(2.8)	Negative	152	(60.6)
Histological type			HER2 IHC		
1	26	(10.4)	3+	232	(92.4)
2	91	(36.3)	2+	19	(7.6)
3	134	(53.4)	HER2 ratio		
Tumor size			Mean	4.8 ± 2.5	
T1	86	(34.3)	HER2 heterogeneity		
T2	139	(55.4)	High (HH)	46	(18.3)
T3 or T4	26	(10.4)	Low (LH)	205	(81.7)
Pretreatment LN metastasis			(Ex-High-Amplification LH)	102	(49.8)
Negative	134	(53.4)	(High-Amplification LH)	103	(50.2)
Positive	117	(46.6)	Stage		
Resected LN metastasis			I	68	(27.1)
Yes	116	(46.2)	II	161	(64.1)
No	135	(53.8)	III	22	(8.8)
Neoadjuvant chemotherapy					
Negative	185	(73.7)			
Positive	66	(26.3)			

**Table 2 cancers-16-01062-t002:** Association between patient characteristics and HER2 heterogeneity.

		HER-2 Heterogeneity		
	Total	High (HH)	Low (LH)	*p*-Value
	*n* = 251, %	*n* = 46, %	*n* = 205, %	
Menopause				
Pre-	96 (38.2%)	23 (50.0%)	73 (35.6%)	0.1
Post-	155 (61.8%)	23 (50.0%)	132 (64.4%)	
Histological type				
IDC	244 (97.2%)	43 (93.5%)	201 (98.0%)	0.118
ILC	7 (2.8%)	3 (6.5%)	4 (2.0%)	
Histological grade				
1	26 (10.4%)	3 (6.5%)	23 (11.2%)	0.421
2	91 (36.3%)	20 (43.5%)	71 (34.6%)	
3	134 (53.4%)	23 (50.0%)	111 (54.1%)	
Tumor size				
T1	86 (34.3%)	14 (30.4%)	72 (35.1%)	0.665
T2–T4	165 (65.7%)	32 (69.6%)	133 (64.9%)	
Pretreatment LN metastasis				
Negative	134 (53.4%)	19 (41.3%)	115 (56.1%)	0.098
Positive	117 (46.6%)	27 (58.7%)	90 (43.9%)	
Resected LN metastasis				
Negative	185 (73.7%)	31 (67.4%)	154 (75.1%)	0.373
Positive	66 (26.3%)	15 (32.6%)	51 (24.9%)	
ER				
Positive	167 (66.5%)	40 (87.0%)	127 (62.0%)	0.002
Negative	84 (33.5%)	6 (13.0%)	78 (38.0%)	
PgR				
Positive	99 (39.4%)	29 (63.0%)	70 (34.1%)	5.46 × 10^−4^
Negative	152 (60.6%)	17 (37.0%)	135 (65.9%)	
HER2 IHC				
3+	232 (92.4%)	40 (87.0%)	192 (93.7%)	0.128
2+	19 (7.6%)	6 (13.0%)	13 (6.3%)	
HER2 ratio				
Mean	4.8 ± 2.5	2.3 ± 0.4	5.4 ± 2.4	1.03 × 10^−43^
Neoadjuvant chemotherapy				
Yes	116 (46.2%)	27 (58.7%)	89 (43.4%)	0.086
No	135 (53.8%)	19 (41.3%)	116 (56.6%)	
Stage				
I	68 (27.1%)	8 (17.4%)	60 (29.3%)	0.094
II	161 (64.1%)	31 (67.4%)	130 (63.4%)	
III	22 (8.8%)	7 (15.2%)	15 (7.3%)	

**Table 3 cancers-16-01062-t003:** Univariate and multivariate analyses of various factors on disease-free survival in HER2-positive patients.

			Univariate Analysis			Multivariate Analysis		
Characteristic	N	Event (N)	HR	95% CI	*p*-Value	HR	95% CI	*p*-Value
cT								
T1	86	8	1	1				
T2–T4	165	22	1.41	0.63–3.16	0.41	1.04	0.45–2.40	0.932
ER								
Positive	167	20	1			1		
Negative	84	10	0.99	0.46–2.12	0.985	1.63	0.62–4.29	0.326
PgR								
Positive	99	14	1			1		
Negative	152	16	0.76	0.37–1.56	0.452	0.95	0.39–2.33	0.918
Pretreatment LN metastasis								
Negative	134	10	1			1		
Positive	117	20	2.33	1.09–4.98	0.029	1.23	0.43–3.52	0.696
Resected LN metastasis								
Negative	185	16	1			1		
Positive	66	14	2.6	1.27–5.32	0.009	2.26	0.81–6.30	0.12
HG								
1 or 2	117	14	1			1		
3	134	16	1.04	0.51–2.13	0.919	1.24	0.59–2.63	0.57
HER2 heterogeneity								
Low (LH)	205	17	1			1		
High (HH)	46	13	3.8	1.84–7.83	2.95 × 10^−4^	4.03	1.83–8.87	5.51 × 10^−4^

**Table 4 cancers-16-01062-t004:** Univariate and multivariate analyses of various factors on overall survival in HER2-positive patients.

			Univariate Analysis			Multivariate Analysis		
Characteristic	N	Event (N)	HR	95% CI	*p*-Value	HR	95% CI	*p*-Value
cT								
T1	86	8	1			1		
T2–T4	165	22	2.22	0.63–7.78	0.214	1.44	0.39–5.24	0.584
ER								
Positive	167	20	1			1		
Negative	84	10	0.67	0.22–2.07	0.485	0.78	0.21–2.94	0.715
PgR								
Positive	99	14	1			1		
Negative	152	16	0.88	0.33–2.36	0.797	1.67	0.53–5.26	0.382
Pretreatment LN metastasis								
Negative	134	10	1			1		
Positive	117	20	4.84	1.38–16.99	0.014	2.79	0.61–12.90	0.188
Resected LN metastasis								
Negative	185	16	1			1		
Positive	66	14	3.71	1.38–9.96	0.009	1.83	0.53–6.26	0.336
HG								
1 or 2	117	14	1			1		
3	134	16	0.7	0.26–1.88	0.482	0.84	0.31–2.33	0.745
HER2 heterogeneity								
Low (LH)	205	17	1			1		
High (HH)	46	13	3.55	1.32–9.54	0.012	3.1	1.06–9.08	0.039

## Data Availability

Data are contained within the article and Appendix A.

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
