# Peer review of "High HER2 Intratumoral Heterogeneity Is a Predictive Factor for Poor Prognosis in Early-Stage and Locally Advanced HER2-Positive Breast Cancer"

_cancers, 2024, doi:10.3390/cancers16051062_

Round 1
Reviewer 1 Report
Comments and Suggestions for Authors
The manuscript concerns intratumoural heterogeneity in Her2 expression in breast cancer as an independent prognostic factor.
The manuscript readable in general but there are some major some syntax errors that probably require a review of the English usage within it.
The findings are interesting and of great potential practical use. The dilemma of Her2-based heterogeneity is a frequent one facing clinical pathologists and oncologists. The decisions made based on the presence of heterogeneity should have a more evidence based approach as they do with the clearer ASCO groups. The approach taken by the authors appears novel, particularly how to recognise high or low heterogeneity. The authors also mention Her2 Low in the discussion as this would seem to be the obvious potential treatment pathway to mimick in Her2 with high heterogeneity.
However, there are some issues with the manuscript as is that require attention before it could be recommended for acceptance in my opinion.
I note the authors use FISH rather than DDISH as the method for describing Her2 heterogeneity using Her2 FISH amplification histograms. Whilst FISH is an acceptable method for assessing the Her2 status of individual cell nuclei it is not as robust as DDISH and image analysis heat maps to assess heterogeneity. The authors should address this methodology question in depth referring to other author’s use of DDISH in this way.
The authors in the text mainly refer to very high or high heterogeneity but in figures can refer to ex-high. I find this somewhat confusing, could the authors be consistent with their use of terms.
In the methods, the authors refer to 252 consecutive patients recruited. In the next paragraph they say all 251 patients received 12 months of adjuvant trastuzumab. They then say that surgery was performed on 122 patients and 116 had neoadjuvant therapy followed by surgery which is 238. What about the other 13 patients. Can the authors clarify their numbers please?
I find Table 1 confusing and it needs reformatting.
The authors state that FISH was calculated for more than (>)20 nuclei but how many more? 20 is the standard count.
They also state that manual or semi-automated protocols were used to assess copy number. This needs more detail. Using a manual method or a semi-automated method for the same sample can result in different values potentially if we don’t know what the detail of the procotols are. Furthermore, are the protocols image analysis based? If so then what software was used, etc. These are vital facts for the reader to know and need to be detailed more than just OCR.
In Table 2 the association between Her2 high and low is not significant in pretreatment LN metastasis (0.098) and this needs to be discussed more in the text as to why this wasn’t significant if Her2 high heterogeneity is associated with poorer prognosis.
It would be interesting if the authors could describe if this phenomenon of low and high heterogeneity is apparent in other cancers such as gastric or endometrial.
The results relate distinctly to pathological features of a breast cancer but there doesn’t appear to be input from pathology on this paper. Issues such as the use of FISH rather than DDISH may have benefited from Pathologist input.
Comments on the Quality of English Language
There are as I mentioned some syntax errors and gaps in the text, for example in the discussion, there is a paragraph on page 11/13 (for me) which ends in “breast cancer. Because in”
Author Response
Thank you very much for your review of our manuscript. We found the comments by the reviewers to be very useful in improving our manuscript. Accordingly, we have revised our manuscript in full compliance with the reviewer's comments. The following describes the changes and responses made:
Reviewer #1:
Reviewer’s comment: I note the authors use FISH rather than DISH as the method for describing Her2 heterogeneity using Her2 FISH amplification histograms. Whilst FISH is an acceptable method for assessing the Her2 status of individual cell nuclei it is not as robust as DISH and image analysis heat maps to assess heterogeneity. The authors should address this methodology question in depth referring to other author’s use of DISH in this way.
Response:
As requested by the reviewer, the following sentence have been modified in the Introduction and in the Discussion.
Dual-color in situ hybridization (DISH) is also a molecular diagnostic technique used to assess the status of the HER2 gene in breast cancer and other malignancies [23] [24] [25]. DISH probes are designed to bind to the HER2 gene and CEP17 and can be visualized under a microscope. DISH allows for the simultaneous detection of both the HER2 gene and CEP17, providing comprehensive information on gene amplification, and allows for quantitative analysis, enabling the precise measurement of gene amplification levels.
Recently, Seol et al. reported that the presence of HER2 heterogeneity associated with decreased disease-free survival (DFS) was treated with only 25% adjuvant trastuzumab therapy [26]. In contrast, Shen et al. revealed that HER2 heterogeneity detected with a HER2 gene protein assay (GPA) was associated with poor overall survival (OS) and increased distal metastasis in patients with HER2-positive breast cancer [21] [27]. In the report, HER2 gene amplification was evaluated with HER2 GPA, which combines HER2 IHC and DISH for simultaneous evaluation at both the protein and gene at the tissue level. However, The DISH method frequently presents with non-specific background deposition, including debris as subtle silver deposits, which elude automated analysis. Furthermore, the DISH method yields manual count-ing dependent on the measurer, requiring training for signal counting and susceptible to inter-measurer variability in interpretation [28] [23].
Similary, Shen et al. reported a significantly poorer prognosis in patients with HH tumors than in those with LH tumors, to be evaluated with HER2 GPA, which combines HER2 IHC (the protein) and DISH (HER2 gene amplification) at the tissue level [27] [21]. In addition, Hou et al. and Horii et al. revealed that HER2 ITH with HER2 GPA method is an independent factor predicting incomplete response to anti-HER2 neoadjuvant chemotherapy [34] [35]. However, HER2 GPA method as well as DISH yields manual counting and demands advanced technical expertise and specialized equipment. Their methods lacks standardized staining and interpretation methods, making it susceptible to false positives influenced by background factors such as debris. Also, the HER2 GPA was not analyzed with the distribution of HER2 gene copy number within a tumor sample.
Reviewer’s comment: The authors in the text mainly refer to very high or high heterogeneity but in figures can refer to ex-high. I find this somewhat confusing, could the authors be consistent with their use of terms.
Response: As requested by the reviewer, the following sentence have been modified in the Result.
Low Heterogeneity tumors were divided into the ex-high amplification LH group (Ex-High LH) (>4.9%) and a high amplification group LH (High LH) (<4.9%) according to the median value of HER2 FISH as the cut-off. high heterogeneity was not divided into very high and high heterogeneity
Reviewer’s comment: In the methods, the authors refer to 252 consecutive patients recruited. In the next paragraph they say all 251 patients received 12 months of adjuvant trastuzumab. They then say that surgery was performed on 122 patients and 116 had neoadjuvant therapy followed by surgery which is 238. What about the other 13 patients. Can the authors clarify their numbers please?
Response:
As requested by the reviewer, the following sentence have been modified in the Materials and Methods (2.1. Patients selection and breast tumor samples).
Materials and Methods has been modified in collect number of patients as below. This study consecutively recruited 252→251 patients with HER2-positive invasive breast carcinoma (mean age: 56.6 years; range: 28–97 years) treated with adjuvant or neoadjuvant HER2-targeted therapies. These patients underwent mastectomy or breast-conserving surgery from February 2009 to January 2018 at Osaka University Hospital, Osaka, Japan. All 251 patients received 12 months of adjuvant trastuzumab. Surgery was performed on 122→185 patients, followed by anti-HER2 adjuvant chemotherapy, and 66→116 patients received neoadjuvant chemotherapy followed by surgery (anthracycline and taxane-based chemotherapy, n = 146; taxane-based chemotherapy, n = 87; anthracycline-chemotherapy, n = 5).
Reviewer’s comment: The authors state that FISH was calculated for more than (>)20 nuclei but how many more? 20 is the standard count. They also state that manual or semi-automated protocols were used to assess copy number. This needs more detail. Using a manual method or a semi-automated method for the same sample can result in different values potentially if we don’t know what the detail of the procotols are. Furthermore, are the protocols image analysis based? If so then what software was used, etc. These are vital facts for the reader to know and need to be detailed more than just OCR
Response: As requested by the reviewer, the following sentence have been modified in the Materials and Methods (2.2. HER2 FISH assay).
All tumor tissues underwent HER2 FISH assay. HER2 status, including HER2 gene signals and ratios, was determined by FISH using the PathVysion HER-2 DNA probe Kit (Abbott Laboratories, Abbott Park, IL). Regarding the kits, the tumor FFPE tissue slides were dried and 20 µL DAPI counterstain solution was applied. Then, a fluorescence microscope was used to capture both orange-fluorescent signals corresponding to HER2/neu and green-fluorescent signals corresponding to CEP17. For the histograms of HER2 FISH, all of the fluorescence images were automatically identified by histograms based on HER2 FISH amplification for the protocol of Duet-3/Setup Station/SOLO2 system (BioView, Rehovot, Israel). Meanwhile, for counting the value of HER2 FISH, a fluorescence microscope was used to count the number of those fluorescent signals under the manual or semi-automated protocols analysis. First, samples were analyzed with semi-automated protocol and then samples failed semi-automated protocol analysis were analyzed with manual protocol. The number of cells counted was more than 60 in semi-automated protocols, and it was 20 in the manual. The ratio between the HER2/neu and the CEP17 signal count was calculated. HER2 was considered positive when a tumor contained more than two genes per cell for the FISH assay of HER2. Experienced molecular pathologists used the 2018 HER2 ASCO/CAP updated guidelines to analyze the HER2 status.
Reviewer’s comment: In Table 2 the association between Her2 high and low is not significant in pretreatment LN metastasis (0.098) and this needs to be discussed more in the text as to why this wasn’t significant if Her2 high heterogeneity is associated with poorer prognosis.
Response: discussion: As requested by the reviewer, the following sentence have been modified in the Discussion.
In Table 2, HH tumors were not significantly associated in pretreatment LN metastasis (0.098). The reason is that the HER2 ITH may contribute to the therapeutic effect (suppression of metastatic recurrence) of HER2 treatment and may has little to do with cancer invasion, metastasis, and the progression to LN metastasis. High proliferative cancer cells in both HH and LH tumors may be susceptible to pretreatment LN metastasis.
Reviewer’s comment: It would be interesting if the authors could describe if this phenomenon of low and high heterogeneity is apparent in other cancers such as gastric or endometrial.
Response: Discussion:As requested by the reviewer, the following sentence have been modified in the Discussion.
Meanwhile, it has been confirmed that HER2-positive gastric cancer responds to HER2-targeted therapy as well [46]. The positivity rate of HER2 in gastric cancer is lower compared to breast cancer, resulting in lower effectiveness of HER2-targeted therapies [47] [48]. Moreover, there is a higher probability of HER2 heterogeneity in gastric cancer compared to breast cancer [49] [50]. Our study might have clinical practice impact that our novel analysis clinically t has the potential to become a new classification of HER2 ITH and predict prognosis in patients with HER2-positive gastric cancer as well as breast cancer, as a discriminative, inexpensive and easy-to perform method.

Reviewer 2 Report
Comments and Suggestions for Authors
Tanie et al., prominent researchers in the field, investigated the impact of HER2 intra-tumoral heterogeneity in both low and high HER2-expressing primary breast tumors following adjuvant HER2-targeted therapies. Utilizing HER2 FISH methodology, the authors classified patients into low- or high-heterogeneity (LH and HH, respectively) groups based on HER2 expression. The analysis revealed that HH patients showed a stronger correlation with HER2 IHC, as well as elevated expression of estrogen and progesterone receptors. Moreover, HH patients exhibited a higher incidence of distant metastasis and poorer survival rates compared to the LH group.
In this manuscript, the authors address a crucial clinical problem and present well-organized content. However, there is room for improvement in conveying the novelty and necessity of the research. I recommended this manuscript for publication after a statement regarding the novelty of this paper in the introduction section and extension of the discussion to highlight the importance of this manuscript in the field.
Author Response
Thank you very much for your review of our manuscript. We found the comments by the reviewers to be very useful in improving our manuscript. Accordingly, we have revised our manuscript in full compliance with the reviewer's comments. The following describes the changes and responses made:
Reviewer #2:
Reviewer ’s comment: There is room for improvement in conveying the novelty and necessity of the research. I recommended this manuscript for publication after a statement regarding the novelty of this paper in the introduction section and extension of the discussion to highlight the importance of this manuscript in the field.
Response: As requested by the reviewer, the following sentence have been modified in the end of Introduction, and Discussion.
It is necessity to develop novel methods capable of analyzing HER2 ITH within cancer tissues automatically, and to make unbiased analyses, rather than relying on manual procedures. Also, to assess more clear-cut heterogeneity of HER2, the novel method is necessary to evaluate the HER2 copy number quantification, as HER2 gene copy numbers are present in each of those cancer cells. In additional, it is desirable that the method proves beneficial in predicting prognosis and anti-HER2 therapies sensitivity. We developed a novel method to confirm the shape of HER2 FISH distributions with HER2 gene copy number within a tumor sample, and analyze HER2 ITH automatically. Also, we conducted our method to investigate the association between HER2-targeted therapy and HER2 heterogeneity. This study defined HER2 ITH using Gaussian mixture modeling (GMM) analysis of histograms based on HER2 FISH distributions. Recently A GMM classifier demonstrated potential use for clinical validation of markers and determination of target populations to improve molecular stratification of patients with breast cancer [32]. We aimed to develop a novel method to analyze HER2 ITH automatically, and determine the clinical association of high HER2 heterogeneity with poor prognosis due to resistance to postoperative adjuvant therapy with HER2-targeted agents in primary breast cancer, through the application of GMM analysis.
The strength of our method is able to analyze HER2 ITH within cancer tissues automatically and to make unbiased analyses of HER2 ITH, rather than relying on manual procedures. Also, our method can evaluate the HER2 copy number quantification within a tumor sample and assess more clear-cut heterogeneity of HER2, because HER2 gene copy numbers are present in each of those cancer cells within a tumor sample.
Besides, another strength of our analysis is an easy, low-cost method only to read histograms of FISH diagnostic reports and perform image processing analysis. The findings of this study provide valuable insights into the efficacy of our novel method of HER2 ITH analysis, and the highlighting is its potential as a clinically useful of predicting prognosis and anti-HER2 therapies sensitivity. And then, the clinical significance of identification of our analysis for the prediction of the prognosis seems to deserve further investigation of multicenter research.

Reviewer 3 Report
Comments and Suggestions for Authors
This manuscript is a retrospective analysis of intratumoural heterogeneity and its impact on the prognosis of patients with HER2-positive early breast cancer treated in a Japanese hospital. In the era of heightened attention to HER2 expression, especially with the advent of new drugs such as novel antibody-drug conjugates like trastuzumab deruxtecan (TDxD), which have blurred the dichotomous definition of HER2 positivity, this topic is of great interest.
The study is well designed and a comprehensive analysis has been conducted which provides intriguing results. Considering that the treatment of HER2-positive early breast cancer is still based on the traditional definition of HER2 positivity, assessing the impact and potential implication of HER2 heterogeneity on treatment is an important issue.
However, improvements are needed to improve the clarity and flow of the manuscript. In addition, the manuscript needs improvement in terms of English language usage and expression.
Here are some suggestions:
The Introduction section needs to be better organised to give a clearer overview of the contextual framework of the study.
- Please provide a more comprehensive definition of tumour heterogeneity, including the concepts of spatial and temporal heterogeneity.
- Please provide a detailed definition of HER2 positivity.
- Recheck the sentence: "Currently, the primary application of HER2 assessment lies in the prediction of responsiveness to anti-HER2 therapy in the neoadjuvant and adjuvant settings". This is not clear.
- Rephrase the paragraph that mentions Seol et al. explaining the relationship between HER2 heterogeneity and adjuvant trastuzumab therapy. The next sentence does not seem to contradict the first. [Recently, Seol et al. reported that the presence of HER2 heterogeneity associated with decreased disease-free survival (DFS) was treated with only 25% adjuvant trastuzumab therapy [21]. In contrast, Shen et al. revealed that HER2 heterogeneity detected with a HER2 gene protein assay (GPA) was associated with poor overall survival (OS) and increased distal metastasis in patients with HER2-positive breast cancer [19][22].]
Consider addressing this issue in the Discussion section:
- Figure 1, introduce Ex-High LH and High-LH groups. Please explain their definitions and the rationale for this subclassification in the main text.
- Consider citing the DAISY study when discussing the methods used to assess HER2 heterogeneity.
- Elaborate on the therapeutic implications of the results. If the authors were to hypothesise an algorithm incorporating tumour heterogeneity, what algorithm would they propose? Given the efficacy of Tdxd in HER2 low (awaiting those of HER2 ultra- low) in the metastatic setting, could tumour heterogeneity be a criterion for assigning this treatment to these patients over standard anti-HER2 therapy at the early stage?
- Assess the relevance of focusing on tumour heterogeneity, particularly in light of drugs such as TdXd that appear to be able to largely overcome this problem (DAISY trial).
Overall, the article is interesting, but it would benefit from some improvements and a thorough revision of the English language.
Comments on the Quality of English LanguageMinor editing of English language required
Author Response
Thank you very much for your review of our manuscript. We found the comments by the reviewers to be very useful in improving our manuscript. Accordingly, we have revised our manuscript in full compliance with the reviewer's comments. The following describes the changes and responses made:
Reviewer #3:
Reviewer’s comment: The Introduction section needs to be better organised to give a clearer overview of the contextual framework of the study.
- 1:Please provide a more comprehensive definition of tumour heterogeneity, including the concepts of spatial and temporal heterogeneity.
- 2:Please provide a detailed definition of HER2 positivity.
Response: As requested by the reviewer, the following sentence have been modified in the Introduction.
- 1:Tumor heterogeneity refers to the diversity of cancer cells within a tumor or among tumors, encompassing variations in genetic mutations, gene expression patterns, and cellular characteristics. This heterogeneity can occur spatially, with different regions of a tumor exhibiting distinct features, and temporally, as cancer cells evolve and adapt over time [2]. [13] [14] .
- 2:The definition of HER2-positive cancer is typically determined primarily immunohistochemistry (IHC) and in situ hybridization (ISH). ISH involves the simultaneous detection of HER2 gene amplification and chromosome 17 (CEP17) to assess HER2 gene status in tumor tissue. If the IHC test reveals a score of 3+, indicating strong HER2 protein expression, or if the IHC test shows a score of 2+ along with confirmed HER2 gene amplification via ISH, the cancer is classified as HER2-positive [12].
Reviewer’s comment: Recheck the sentence: "Currently, the primary application of HER2 assessment lies in the prediction of responsiveness to anti-HER2 therapy in the neoadjuvant and adjuvant settings". This is not clear.
Response: As requested by the reviewer, the following sentence have been modified in the Introduction.
The primary application of HER2 assessment lies in the improvement for patient survival to anti-HER2 therapy in the neoadjuvant and adjuvant settings
Reviewer’s comment: Consider addressing this issue in the Discussion section:
- Figure 1, introduce Ex-High LH and High-LH groups. Please explain their definitions and the rationale for this subclassification in the main text.
Response: As requested by the reviewer, the following sentence have been modified in the Discussion.
In additional, we investigated whether the level of HER2 FISH amplification is associated with the prognosis of LH, and LH tumors were divided into the ex-high amplification LH group (Ex-High LH) and a high amplification group LH (High LH). As a result, in HER2-positive LH tumors, our study revealed that the ratio of HER2 FISH (mean number of HER2 FISH signals/cell) (Ex-High LH vs. High LH) was not associated with prognosis of DFS rates and OS rates (Fig. 2C and 2D).
Reviewer’s comment: - Consider citing the DAISY study when discussing the methods used to assess HER2 heterogeneity.
- Elaborate on the therapeutic implications of the results. If the authors were to hypothesise an algorithm incorporating tumour heterogeneity, what algorithm would they propose? Given the efficacy of Tdxd in HER2 low (awaiting those of HER2 ultra- low) in the metastatic setting, could tumour heterogeneity be a criterion for assigning this treatment to these patients over standard anti-HER2 therapy at the early stage?
- Assess the relevance of focusing on tumour heterogeneity, particularly in light of drugs such as Tdxd that appear to be able to largely overcome this problem (DAISY trial).
Response: As requested by the reviewer, the following sentence have been modified in the Discussion.
Furthermore, our analysis has the potential to accurately assess ITH not only in HER2-positive breast cancers but also in HER2-low. It is hypothesized that HER2-low breast cancers commonly represent more heterogeneous population than HER2-positive breast cancers [41]. Recently, trastuzumab deruxtecan (T-DXd) treatment resulted in significantly longer relapse-free survival and OS than chemotherapy among patients with HER2-low metastatic breast cancer [42]. This study reported that HER2-low tumors constituted a heterogeneous population that varied in prognosis and sensitivity to systemic treatments. T-DXd, effectively targets tumor cells that express low levels of HER2 and can deliver a potent cytotoxic payload through the bystander effect to neighboring tumor cells heterogeneously expressing HER2, unlike many other approved HER2-targeted therapies [43] [44]. Therefore, in the HER2-low as well as HER2-positive, highly heterogeneous breast cancers might be responsive to treatment of T-DXd that exhibits potent therapeutic efficacy through the delivery of cytotoxic payload. In addition, Mosele et. al. reported that HER2 is a determinant of sensitivity to T-DXd, although modest anti-tumor activity was also observed in a small subset of patients whose cancer did not express HER2 [45]. The study revealed that T-DXd anti-tumor activity increased when HER2 expression was high, modest anti-tumor activity was also observed in patients with HER2 IHC 0, and suggested that very low levels of HER2 could allow uptake of T-DXd and/or that partially mediated drug efficacy. That is to say, the level of HER2 IHC is extremely low or absent, which cannot have implications for T-DXd treatment decisions, and our HER2 FISH distributions analysis without HER2 IHC, might accurately identify the heterogeneity of HER2-low breast or absent, and predict T-DXd efficacy of breast cancers. Hence, our analysis may be useful for predicting the effectiveness of not only standard HER2-targeted therapies but also of the new drug against highly heterogeneous tumors. In the future, we are going to investigate whether our method our analysis can accurately identify the heterogeneity of HER2-low breast cancers or not.

Round 2
Reviewer 3 Report
Comments and Suggestions for Authors
-